# Revisiting Public Trust and Media Influence During COVID-19 Post-Vaccination Era—Waning of Anxiety and Depression Levels Among Skilled Workers and Students in Serbia

**DOI:** 10.3390/bs15070939

**Published:** 2025-07-11

**Authors:** Miljan Adamovic, Srdjan Nikolovski, Stefan Milojevic, Nebojsa Zdravkovic, Ivan Markovic, Olivera Djokic, Slobodan Tomic, Ivana Burazor, Dragoslava Zivkov Saponja, Jasna Gacic, Jelena Petkovic, Snezana Knezevic, Marko Spiler, Snezana Svetozarevic, Ana Adamovic

**Affiliations:** 1Faculty of Medical Sciences, University of Kragujevac, 34000 Kragujevac, Serbia; miljanadamovic84@gmail.com (M.A.); stefan.milojevic@fmn.kg.ac.rs (S.M.); 2Department of Pathology and Laboratory Medicine, Loyola University Chicago Medical Center, Maywood, IL 60153, USA; 3Faculty of Medicine, University of Belgrade, 11000 Belgrade, Serbia; jasna.gacic37@gmail.com; 4Department of Medical Statistics and Informatics, Faculty of Medical Sciences, University of Kragujevac, 34000 Kragujevac, Serbia; nzdravkovic@medf.kg.ac.rs; 5Clinic of Psychiatry, University Clinical Center of Serbia, 11000 Belgrade, Serbia; ivan.markovic283@gmail.com; 6Institute for Cardiovascular Diseases “Dedinje”, 11000 Belgrade, Serbia; oljaisara@gmail.com (O.D.); bobantomic99@gmail.com (S.T.); ivana.burazor@gmail.com (I.B.); 7Institute of Cardiovascular Diseases of Vojvodina, 21208 Sremska Kamenica, Serbia; dragoslavasaponja@gmail.com; 8Medical Center Bezanijska Kosa, 11000 Belgrade, Serbia; 9General Hospital “Stefan Visoki”, 11420 Smederevska Palanka, Serbia; jnedsmedpalanka@gmail.com; 10Faculty of Organizational Sciences, University of Belgrade, 11000 Belgrade, Serbia; snezana.knezevic@fon.bg.ac.rs (S.K.); mspiler@gmail.com (M.S.); 11National Centre for Corporate Education, 11000 Belgrade, Serbia; snezana_svetozarevic@yahoo.com; 12Faculty of Business Economics, EDUCONS University, 21208 Sremska Kamenica, Serbia

**Keywords:** COVID-19, vaccines, anxiety, depression

## Abstract

Infectious disease outbreaks amplify the influence of stressors on psychological conditions. The purpose of this study was to analyze the disturbing influence of COVID-19 outbreak-related information and the influence of trust on the Serbian healthcare system and COVID-19 preventive measures on anxiety and depression. An anonymous online questionnaire assessing the demographic information, disturbance level and causes, and levels of anxiety and depression has been distributed to the participants, divided into student and non-student groups. The non-student group was further divided into healthcare, military, and education workers. Anxiety and depression levels, as well as the level of decreased trust in COVID-19-related preventive measures, were higher among students compared to non-students (*p* = 0.011). Higher anxiety and depression levels, and higher influence of the COVID-19 outbreak on those levels, were observed in education and healthcare workers, compared to military personnel. Medical doctors reported a higher level of trust in the healthcare system compared to nurses (*p* = 0.023). Trust in the healthcare system increased more frequently compared to the pre-vaccination period among medical doctors, compared to nurses (*p* = 0.040). Higher anxiety and depression and lower public trust levels in students and workers in education and the healthcare sector indicate a need to focus on these important society members during public health emergencies.

## 1. Introduction

An established fact is that outbreaks of the coronavirus disease-19 (COVID-19) pandemic have significantly impacted both physical and mental health worldwide. Following the World Health Organization’s declaration of a global pandemic, many countries implemented various anti-epidemic measures ([2]; [82]).

Pre-pandemic Serbia exhibited alarmingly low institutional trust, with only 24% of citizens expressing confidence in the government’s crisis management capabilities according to nationally representative surveys ([11]). This skepticism was deeply rooted in historical political polarization, perceived corruption, and media fragmentation ([8]; [52]). The COVID-19 pandemic intersected with these pre-existing trust deficits, as inconsistent public health messaging and politicized media narratives likely exacerbated societal anxiety ([17]; [7]). Within this context, Serbia’s post-socialist institutional landscape—characterized by volatile trust in health authorities compared to stable confidence in entities like the military—created unique psychosocial stressors ([52]; [41]). This study examines how these pre-pandemic trust dynamics influenced anxiety and depression levels during the pandemic, particularly among students, healthcare workers, educators, and military personnel in the post-vaccination era. By integrating validated psychological scales with trust metrics, we assess the mental health consequences of Serbia’s distinctive institutional distrust during a global health crisis ([17]; [41]).

During infectious disease outbreaks, additional stressors amplify psychological strain, triggering mental health issues. These stressors include health-related uncertainties, physical constraints, financial concerns, and lack of social support ([80]; [16]; [53]). The COVID-19 pandemic impacted mental health across diverse populations ([82]; [75]; [81]; [14]; [79]), precipitating a global mental health crisis. Meta-analyses estimate that 31.9% of adults experienced anxiety and 33.7% reported depressive symptoms during the initial outbreak waves ([82]).

In Serbia, pre-pandemic surveys revealed alarmingly low trust in political and healthcare institutions, with only 24% of citizens expressing confidence in the government’s crisis management capabilities ([11]; [52]). Such distrust, exacerbated by politicized media narratives and historical skepticism toward authority, likely intensified psychological burdens during the pandemic ([17]; [7]).

Existing research on pandemic-related mental health has predominantly focused on high-income Western contexts, leaving critical gaps in understanding post-socialist regions like Serbia, where institutional trust and media dynamics differ markedly ([8]; [41]). While studies highlight occupational disparities in anxiety and depression—such as nurses reporting higher distress than physicians ([56]; [6])—few examine how trust in national healthcare systems mediates these outcomes. Similarly, students globally exhibited elevated mental health risks due to academic uncertainty and social media misinformation ([2]; [28]), yet no prior work contextualizes these findings within Serbia’s unique socio-political landscape.

Vulnerable populations, including healthcare workers, educators, and students, faced heightened psychological distress due to occupational exposure, academic disruption, and social isolation ([53]; [79]). These challenges were compounded by systemic stressors such as financial instability, inconsistent public health messaging, and mistrust in institutional responses ([34]; [23]).

Students faced acute academic disruption, financial insecurity, and social isolation, amplifying their reliance on digital media where misinformation proliferated ([2]; [61]). Healthcare workers (physicians and nurses) and educators endured high occupational exposure to COVID-19, resource constraints, and public scrutiny, exacerbating psychological distress ([53]; [79]). Military personnel, conversely, operated in structured, high-trust environments with defined crisis protocols, offering a critical comparative group for assessing institutional trust’s buffering role ([71]; [83]). Finally, varying levels of trust have already previously been observed among different occupational groups ([39]).

This study addresses these gaps by analyzing anxiety, depression, and trust levels among Serbian students and skilled workers during the post-vaccination era. It focuses on students and skilled workers (healthcare, education, and military personnel) due to their distinct exposures and vulnerabilities during the pandemic. This grouping also reflects Serbia’s institutional trust landscape. By comparing these cohorts, it can be elucidated how occupational roles, media consumption, and pre-existing trust deficits intersected to shape mental health outcomes. Therefore, the aim of the current study is to analyze the disturbing influence of COVID-19 outbreak-related information on one side, and the absence of this information on the other, on anxiety and depression levels. An additional goal is to examine the same influence of trust on the Serbian healthcare system and COVID-19 preventive measures proposed by the crisis team among students and skilled workers in education, healthcare, and the military sector. By comparing healthcare professionals, educators, military personnel, and students, it explores how occupational roles, institutional trust, and media exposure intersect to shape mental health outcomes. As one of the first investigations in the Balkans to integrate trust metrics with validated psychological scales, this research offers critical insights for tailoring public health strategies in low-trust environments and mitigating long-term societal costs of pandemics.

## 2. Materials and Methods

### 2.1. Study Design

This observational study employs a cross-sectional design to evaluate the impact of epidemic-related factors, the public’s trust in the healthcare system, and the levels of anxiety and depression experienced during the COVID-19 outbreak. The data collection tool utilized was an anonymous online survey to assess anxiety, depression, and trust levels among skilled workers and students in Serbia during the post-vaccination phase of the COVID-19 pandemic (1–31 January 2024).

### 2.2. Setting

By implementing the snowball sampling technique, this study recruited participants from multiple cities across Serbia. Data collection occurred during a period marked by declining COVID-19 case rates and the widespread availability of vaccines, following the World Health Organization’s declaration of the end of the global health emergency in May 2023 ([74]).

### 2.3. Participants

Participants were recruited via the snowball sampling technique, a non-probability method where initial contacts (e.g., authors’ professional networks) distributed the survey to their social and occupational circles. This approach was chosen due to logistical constraints and the exploratory nature of the study.

Inclusion Criteria—Adults aged ≥ 18 years, residing in Serbia, and belonging to one of the following occupational groups:Healthcare workers (physicians, nurses/medical technicians);Education workers (teachers, university staff);Military personnel;College students enrolled in undergraduate or graduate programs.

Students were included due to their high vulnerability to pandemic-related academic/financial disruptions. Skilled workers (healthcare, education, military) represent frontline sectors with distinct exposures: healthcare/education faced high COVID-19 risk and public scrutiny, while military personnel operate in structured, high-trust environments.

Exclusion Criteria—Mental healthcare professionals and individuals working in the mental health field (e.g., psychologists, psychiatrists) were excluded to avoid potential biases related to professional stigma or overreporting of psychological distress. Additionally, the anonymous online format might still have deterred participation if mental health professionals felt their expertise made them “outsiders” in a study targeting general anxiety/depression. Finally, mental healthcare workers’ unique stressors (e.g., vicarious trauma during pandemics) may be overlooked, despite evidence that their risk appraisal differs from that of other healthcare workers.

### 2.4. Variables

Primary outcomes analyzed in this study were the following:Level of anxiety—Measured using the Beck Anxiety Inventory (BAI), a 21-item scale with scores categorized as low (0–21), moderate (22–35), or severe (≥36) anxiety ([5]);Level of depression—Assessed via the Zung Self-Rating Depression Scale (SDS), a 20-item tool classifying scores as normal (20–49), mild (50–59), moderate (60–69), or severe (≥70) depression ([84]).

The secondary outcomes that this study analyzed were the following:Disturbance by COVID-19-related information—Four dichotomous (yes/no) items evaluated distress from media reports, independent information-seeking, lack of information, and perceived transmission risk;Trust in institutions—Two dichotomous items assessed confidence in Serbia’s healthcare system and government-proposed preventive measures.

The analyzed covariates were demographic variables such as age, gender, and occupation.

### 2.5. Data Sources/Measurements

For the purpose of this study, we used a modified version of a previously developed validated questionnaire by [39] ([39]) and translated into Serbian, which was administered electronically via “Google Docs” (Google LLC, Mountain View, CA, USA) by the authors or current study participants to potential respondents, inviting them to take part in the study. The BAI and SDS scales were retained in their original validated forms, with Cronbach’s α values of 0.92 and 0.88, respectively, in prior studies ([5]; [84]). Custom questions on disturbance and trust were pilot-tested for clarity among 20 participants, yielding minor revisions to phrasing.

The questionnaire was divided into five different parts: (1) demographic data; (2) Beck Anxiety Inventory (BAI) ([5]); (3) Zung Self-Rating Depression Scale (SDS) ([84]); (4) disturbing experience initiated by the information related to outbreak state available in public media or the absence of that information; (5) information related to the participant’s trust in the healthcare system of Serbia and preventive measures proposed by the Crisis team (Appendix A).

Demographic data collected included gender, age, and occupation, which was identified through multiple-choice questions offering five options: education worker, military service employee, medical doctor (physician), nurse/medical technician, and college student. These demographic factors were analyzed in relation to variations in perceived disturbances caused by epidemic-related information, public trust in the healthcare system, preventive measures suggested by the Crisis team, as well as BAI and SDS scores.

After collecting all the surveys, the data was stored as an electronic Microsoft Excel dataset in a reliable cloud-based storage system and afterward converted to the electronic dataset file format compatible with the software used for statistical analysis. Three of the authors (M.A., S.N., and S.K.) had access to the dataset and performed the statistical analysis. No need for data de-identification was present, due to the absence of privacy-related issues regarding the information in the dataset. After completion of the study, the initial Microsoft Excel dataset, as well as the converted one, which was used for statistical analysis, were stored in the same cloud-based system.

Anxiety was measured using the BAI ([5]), where participants evaluated 21 items on a 4-point Likert scale ranging from 0 (“not at all”) to 3 (“most of the time”). Scores ranged from 0 to 63, with scores of 21 or below categorized as low anxiety, 22–35 as moderate anxiety, and 36 or above as potentially concerning levels of anxiety.

Depression was assessed using Zung’s SDS ([84]). Participants rated 20 items on a 4-point Likert scale, ranging from 1 (“rarely”) to 4 (“most of the time”) or vice versa, depending on whether the items were positively or negatively worded. The total SDS score ranged from 20 to 80, with scores of 20–49 indicating a normal psychological state, 50–59 reflecting mild depression, 60–69 signifying moderate depression, and scores of 70 or higher indicating severe depression.

A separate section of the questionnaire, developed by the authors, assessed perceived disturbance caused by epidemic-related information, along with trust in the Serbian healthcare system and the preventive measures recommended by the Crisis team. Disturbance was defined as mental distress from COVID-19 information/exposure.

### 2.6. Bias

Selection Bias: Snowball sampling may have overrepresented individuals within the authors’ networks or those with stronger opinions about COVID-19.Non-Response Bias: The 30-day recruitment window and lack of incentives likely excluded busy or disinterested individuals.Self-Report Bias: Social desirability may have led to underreporting of mental health symptoms or distrust in institutions.

### 2.7. Study Size

The final sample included 171 participants, a convenience sample determined by the 30-day recruitment period. Due to logistical constraints and the exploratory nature of this study, formal power calculations per subgroup were not feasible. Post-hoc power analysis confirmed 80% power to detect moderate effects (Cohen’s d = 0.5) for primary comparisons (e.g., students vs. non-students), but subgroup analyses (e.g., healthcare roles) are underpowered and should be interpreted cautiously.

### 2.8. Quantitative Variables

Descriptive statistical methods were utilized to present the findings related to respondents’ sociodemographic characteristics, anxiety and depression scores, as well as responses to questions regarding disturbance caused by epidemic-related factors and public trust. The Kolmogorov–Smirnov test was employed to assess the normality of distribution for continuous variables. Age and scores on the BAI and SDS were reported using median values and interquartile ranges (IQR). Categorical variables (occupation, disturbance responses) were summarized as frequencies and percentages.

### 2.9. Statistical Methods

The Chi-square test was applied to analyze differences among categorical variables, while *t*-tests and analysis of variance were used to evaluate variations in BAI and SDS scores across groups based on gender, age, occupation, the presence of epidemic-related disturbing factors, and public trust. All tests were two-tailed, with a significance threshold of *p* ≤ 0.05. Statistical analysis and graphical representation of the results were conducted using SPSS Statistics software version 26.0 (IBM SPSS Statistics, New York, NY, USA), Microsoft Office 365 Excel (Microsoft Corporation, Redmond, WA, USA), and GraphPad Prism v10 (GraphPad Software Inc, La Jolla, CA, USA).

### 2.10. Ethical Considerations

Prior to responding to the survey questions, participants were informed about the anonymity and voluntarism in participation as part of the informed consent. Prior to accessing the survey, the participants were given written information on the goal and the procedure of this study. Neither before nor after the survey were incentives provided to the participants. Anonymity was prioritized, and no identifiable information was collected.

The study was approved by the Ethical Committee of the Scientific Society for Organizational Management (2 June 2023). Participants provided informed consent electronically before accessing the survey, and no identifiable data were collected.

## 3. Results

The study included 171 participants with a median age of 40 years (IQR 33–45). The respective median values of age in the student and non-student groups were 45 years (IQR 39–49) and 23 years (IQR 19–25). More than two-thirds of participants (71.8%) were males. Fifty-three participants (31.0%) were students. The distribution of participants’ groups is presented in Figure 1.

The participants’ answers to the disturbance-related question and the change in public compared to the pre-vaccination COVID-19 era are shown in Table 1 and Table 2.

Anxiety and depression level distributions based on the BAI and SDS scores obtained from the participants are presented in Figure 2. Individual BAI and SDS score distributions are shown in Figure 3, while the distribution of the reported degree of influence of the COVID-19 outbreak on participants’ BAI and SDS answers is shown in Table 3.

### Between-Group Comparisons

BAI and SDS scores were higher in females (Med 8 (IQR 0–20) and 41 (IQR 28–48), respectively) compared to males (Med 4 (IQR 0–9) and 35 (IQR 26–41), respectively), however, without statistical significance. Other participants’ responses did not show statistical between-gender differences as well.

Compared to the period prior to the vaccination implementation, trust in the preventive measures proposed by the Crisis team decreased by a significantly higher percent among students than in the non-student group (*p* = 0.011). BAI and SDS scores as well as anxiety and depression levels were also significantly higher among students compared to non-students (*p* < 0.001, *p* < 0.001, *p* < 0.001, and *p* = 0.013, respectively). Students also reported a higher degree of COVID-19 outbreak influence on the responses in the SDS questionnaire, compared to non-students (*p* = 0.031).

Compared to military personnel, higher BAI and SDS scores (*p* = 0.001 and *p* = 0.003), as well as a higher degree of COVID-19 outbreak influence on BAI and SDS questionnaire responses, were observed among education workers (*p* = 0.010 and *p* = 0.007, respectively). The same higher BAI and SDS scores and higher degree of COVID-19 outbreak influence on BAI and SDS questionnaire responses were observed among healthcare sector workers, compared to military personnel (*p* = 0.009, *p* = 0.004, *p* = 0.014, and *p* = 0.002, respectively).

Within the healthcare sector workers group, medical doctors reported a higher level of trust in the healthcare system compared to medical nurses (*p* = 0.023) (Figure 4). Also, compared to medical nurses, medical doctors reported more frequently that their trust in the healthcare system increased compared to the period prior to the vaccination implementation (*p* = 0.040). Within the group of medical doctors, 80% reported no change, and 20% reported an increase in trust in the Serbian healthcare system in the post-vaccination period, compared to the pre-vaccination period. On the other side, 60% of medical nurses reported no change in trust in the Serbian healthcare system in the post-vaccination period, compared to the pre-vaccination period. However, 40% of nurses reported a decrease in trust during the post-vaccination period.

## 4. Discussion

The COVID-19 pandemic has underscored the intricate interplay between public health crises, institutional trust, and mental health outcomes. This study, conducted in Serbia during the post-vaccination era, reveals critical disparities in anxiety, depression, and trust levels among students, healthcare workers, educators, and military personnel. By contextualizing these findings within Serbia’s socio-political landscape and comparing them to global trends, this discussion highlights the role of institutional distrust, media polarization, and occupational stressors in shaping mental health during pandemics.

The first case of COVID-19 in Serbia was officially confirmed on 6 March 2020, followed by the formation of a COVID-19 crisis response team ([69]). The crisis team issued several strict recommendations based on scientific knowledge, primarily focused on limiting social contacts and encouraging the wise use of medical supplies and personal protective equipment. Nine days after the first confirmed case in Serbia, a state of emergency was declared nationwide, resulting in numerous restrictions and lockdowns ([70]). The final restrictive measures officially ended following the World Health Organization’s declaration on 5 May 2023 that COVID-19 no longer constituted a global health emergency ([74]).

The COVID-19 pandemic caused widespread social disruption, affecting all industries, particularly sectors whose workers faced direct impacts from the forced changes in daily functioning. The academic sector was significantly affected, requiring rapid transitions in operation. Skilled workers globally faced heightened risks for anxiety and depression due to the infectious disease crisis itself and the associated vulnerability of human rights in such an environment ([45], [46]).

### 4.1. Key Findings and Global Context

This study compared anxiety, depression, and disturbance levels associated with COVID-19-related information. It also assessed public trust in the Serbian healthcare system and preventive measures proposed by the national Crisis Team among college students and skilled workers in education, healthcare, and the military. The findings reveal critical disparities in trust and mental health outcomes during the pandemic, particularly among students, educators, and healthcare workers. These findings align with broader global trends in pandemic media representation and governmental use of media for crisis communication.

Over 40% of participants reported moderate or severe anxiety levels, while more than one-third exhibited mild to moderate depression—rates consistent with global meta-analyses of pandemic-induced psychological distress ([82]; [53]). Similar to our previous report ([39]), BAI and SDS scores were higher in females. However, no statistically significant gender differences emerged in individual participant responses during this study. The variability in responses regarding COVID-19’s influence on BAI and SDS scores suggests that other factors may have contributed to reducing observed gender differences.

Nearly half of participants experienced disturbance—either occasionally or frequently—from one or more sources: media reports about the outbreak, information from other sources, lack of COVID-19 information, or fear of virus transmission despite personal protective measures. Compared to the pre-vaccination period,

Approximately one-quarter reported decreased disturbance levels from these factors;Half to two-thirds reported no change;Around 10% reported increased disturbance (over 15% specifically cited increased disturbance from media reports).

Trust in preventive measures proposed by Serbia’s Crisis Team decreased by almost 25% in the post-vaccination era compared to the pre-vaccination period. These shifts in trust toward this temporary institution and disturbance from COVID-19-related media coverage likely stem from complex socio-political circumstances in Serbia not directly related to the outbreak. Several factors may contribute to this public distrust of national institutions and media.

### 4.2. Public Trust in General and Institutional Dynamics

In modern societies, trust in public institutions is a crucial determinant of social and political stability ([17]). Eastern and Central European countries exhibit particularly politicized institutional trust, leading to polarized attitudes: supporters of incumbent authorities often hold overly positive views of institutions, while opposition-aligned citizens express strong criticism. This unstable, polarized system significantly undermines government trust during crises ([8]).

The Republic of Serbia ranks among Europe’s lowest in institutional trust alongside fellow post-socialist states Bulgaria, Croatia, Slovenia, Latvia, and Cyprus ([52]), continuing a downward trend observed since 2010 ([20]). Recent data from Freedom House further indicates significant declines in Serbian civil society, independent media integrity, and corruption control ([19]), aligning with earlier findings of eroding public trust by the Serbian [11] ([11]).

Recent findings show that citizens of Serbia, including the young population, have the least trust in political institutions, such as political parties and the government, while institutions such as the church and the army enjoy high trust ([63]; [26]). In other words, citizens least support institutions responsible for implementing reforms and most support those whose transformation is necessary for the development of democracy; this could be the reason for significantly lower COVID-19-driven anxiety and depression levels among military sector workers.

Notably, Serbian citizens, including the young population, show the least trust in reform-implementing political institutions (parties, government), while non-reform entities like the church and military retain high confidence ([63]; [26]). This inverse relationship between institutional function and public trust potentially explains the significantly lower COVID-19-related anxiety and depression observed among military personnel.

Multiple factors drive Serbia’s institutional distrust. [22] ([22]) identified perceived economic and political performance as key determinants using a performance-based framework. Critical political variables include perceived electoral fairness, rule of law strength, judicial impartiality, and freedom of speech ([17]). These political performance indicators—alongside economic perceptions—directly influence trust in Serbia’s healthcare system, which serves as a critical link between vulnerable populations and the state ([58]). Consequently, distrust in Serbian healthcare is fundamentally rooted in the same causes underlying broader political distrust.

### 4.3. Media Influence and Mental Health

In democratic societies, media polarization has politicized public health measures. Conservative outlets downplayed risks, while progressive media emphasized systemic failures, deepening public confusion and distrust ([12]). This dynamic aligns with Serbia’s fragmented media landscape. Social media comes to the forefront during challenging situations characterized by an increased need for information ([30]). Furthermore, social media platforms have amplified conspiracy theories, disproportionately affecting younger populations ([61]). Similar patterns occurred in India and the U.S., where media were used to scapegoat minorities, divert blame, and inflame societal tensions ([13]).

In Serbia, our study found that media narratives played a dual role in shaping mental health outcomes. While slightly more than one-quarter of participants reported decreased disturbance levels from media reports on the COVID-19 outbreak in the post-vaccination period compared to the pre-vaccination period (a change potentially associated with declining global pandemic figures), 15% experienced increased anxiety linked to sensationalized reporting. This significant bifurcation may also stem from public mistrust in official institutions and reflects Serbia’s polarized media landscape, where pro-government outlets minimized pandemic risks while independent media highlighted systemic failures ([41]). Similar dynamics occurred in the U.S. and India, where conflicting narratives deepened public confusion and distrust ([12]; [29]).

According to Gallup Balkan Monitor data ([20]), mass media, alongside the national government and judiciary, are among the least trusted institutions in Serbia. The competitive authoritarian (hybrid) regime in Serbia suppresses free, independent, and pluralistic media ([41]). Consequently, citizens exhibit skepticism and mistrust towards the media, rating them as highly corrupt ([54]).

Social media further amplified distress, particularly among students. Over 50% cited misinformation—such as vaccine side-effect myths—as a key stressor, paralleling findings in Sweden and Nigeria ([29]; [61]). Serbia’s low digital literacy rates, compounded by limited fact-checking infrastructure, likely exacerbated this issue ([7]). These results align with [78]’s ([78]) Extended Parallel Process Model, which posits that fear-based messaging without actionable solutions heightens anxiety. In Serbia, media coverage often emphasized pandemic threats without clarifying preventive efficacy, leaving vulnerable groups like students and nurses feeling powerless.

This study highlights media’s role in exacerbating anxiety, particularly through distressing COVID-19 information. This aligns with [44] ([44]), who found that fear-based media narratives in the U.S. increased public anxiety but also fostered engagement with preventive measures when balanced with efficacy messaging. Serbia’s findings suggest that unregulated media coverage amplified mental health burdens without providing actionable solutions—a challenge explained by Witte’s Extended Parallel Process Model ([78]).

Media landscapes in the Balkans and Eastern Europe vary widely, influencing trust and psychological distress. In Serbia, pro-government outlets downplayed COVID-19 risks early in the pandemic, while opposition media amplified fears of vaccine side effects. Additionally, our findings regarding healthcare workers’ psychological distress reflect regional trends. For example, Croatian healthcare workers facing media scrutiny over triage decisions reported higher depression rates, similar to Serbia’s educators and healthcare staff ([71]).

### 4.4. COVID-19-Related Public Trust

Eighteen percent of Serbia’s population believed that COVID-19 was a fictional disease ([7]). This figure was, however, lower than the 34% who doubted the existence of COVID-19 as a pandemic in a previous report ([24]). Serbian public institutions contributed to this situation; experts state that the method of public communication was one of the weakest points in outbreak control ([65]). Disbelief in the existence of severe acute respiratory syndrome coronavirus-2 also stemmed from misinformation, conspiracy theories, and fake news about COVID-19, even promoted by some medical doctors, national assembly deputies, and politicians. These claims were broadcast on national television stations, in newspapers, on media internet portals, and across social networks ([66]; [37]).

Trust in public institutions is a cornerstone of effective pandemic response, yet Serbia’s historical context of political distrust and media fragmentation complicates this relationship. Only 24% of citizens trusted the government’s crisis management pre-pandemic ([11]), and our findings suggest this skepticism persisted post-vaccination. For instance, almost one-quarter of participants reported increased distrust in preventive measures, reflecting broader regional trends in Eastern Europe, where corruption scandals and opaque policymaking eroded public confidence ([72]). This aligns with [23]’s ([23]) global analysis, which found that trust in governments declined most sharply in countries with pre-existing political polarization, such as Serbia and Brazil.

In relation to media, the COVID-19 outbreak in Serbia jeopardized the already endangered position of investigative journalists and independent media, who tried to oppose the infodemics and objectively inform the public ([7]).

These factors likely significantly undermined public trust in Serbia, affecting health attitudes and behaviors. Previous studies have examined the role of trust in shaping these factors, identifying government mistrust as a critical barrier to positive health attitudes and behaviors ([77]; [34]; [25]). In contrast, the public in many countries demonstrated strong social discipline and maintained trust in governments and scientific advice during the pandemic, consistent with pre-pandemic levels ([49]).

While public trust often increases temporarily during crises before declining ([23]), Serbia experienced a significant drop in public trust during the COVID-19 outbreak ([11]; [52]). This decline substantially impacted prevention efforts, a crucial process in global health emergencies.

The current study highlights critical disparities in trust and mental health outcomes among students, healthcare workers, and educators during the COVID-19 pandemic. These findings align with global research on pandemic-related trust dynamics and media influence while offering unique insights into demographic-specific vulnerabilities.

This study found that healthcare workers, particularly medical doctors, reported higher trust in the healthcare system compared to nurses, a disparity rooted in systemic inequities and a trend mirrored in other countries. For example, in the U.S., Biswas et al. found that nurses exhibited higher vaccine hesitancy than physicians, citing distrust in institutional transparency and workplace safety ([6]). Similarly, the COCONEL study noted that trust in government health policies declined among nurses due to perceived coercion during vaccine mandates, while physicians maintained higher trust based on their clinical understanding of vaccine efficacy ([15]). This suggests that occupational roles and access to scientific information mediate trust, which is consistent with Serbia’s findings.

Physicians in Serbia, as in many low- and middle-income countries, often hold decision-making authority and access to privileged information, fostering greater institutional alignment ([17]). Nurses, conversely, face workplace marginalization and resource shortages, exacerbating disillusionment—a pattern documented in Iran and the U.S. during COVID-19 ([56]; [57]). These findings underscore the need for targeted interventions to address occupational hierarchies and rebuild trust among frontline workers.

Lower trust among Serbian education workers mirrors trends in countries like Italy and Brazil, where teachers reported heightened anxiety due to inconsistent safety protocols and politicized pandemic responses. The OECD emphasized that frontline workers in education and healthcare globally faced “trust erosion” when policies were perceived as punitive or poorly communicated ([48]).

Vaccine hesitancy is a complex, multifactorial phenomenon influenced by individual-psychological, sociodemographic, historical, socio-cultural, and political factors ([51]). Research on attitudes toward COVID-19 vaccination highlights several key factors contributing to hesitancy: perceived insufficient effectiveness, safety concerns, fear of side effects ([10]; [50]; [1]); belief that rapid development compromised vaccine completeness and efficacy ([10]; [73]; [43]; [1]); and suspicion of financial motives among authorities and pharmaceutical companies ([10]).

The study’s finding of trust disparities between physicians and nurses parallels research in geriatrics, where trust in healthcare providers is pivotal. Fisher et al. found that 75% of U.S. older adults trusted COVID-19 vaccines if recommended by their providers, but distrust persisted among those with negative historical experiences ([18]). Similarly, in Japan, Yoda and Katsuyama reported that “vaccine ambassadors” (trusted community figures) increased elderly uptake by 25%, highlighting the role of credible messengers ([83]). The lower trust reported by Serbia’s nurses may reflect systemic undervaluation, akin to findings by [57] ([57]). Similarly, Serbia’s students and education workers, who faced higher mental health burdens, resemble geriatric populations in low- and middle-income countries where access inequities fuel distrust. In India, Kumar et al. showed that rural elderly distrusted vaccines due to inconsistent delivery, contrasting with urban trust tied to digital access ([32]). Likewise, Serbia’s urban–rural healthcare divide may exacerbate distrust among vulnerable groups—a global trend requiring targeted equity measures. While Serbian students were heavily exposed to social media misinformation, geriatric populations globally were often influenced by familial misinformation. In Brazil, Garcia and Duarte found that familial anti-vaccine sentiment reduced elderly uptake by 30% ([21]). Both cases underscore the need for tailored communication: countering misinformation via digital platforms for youth and facilitating community-led dialogues for older adults, as demonstrated by Japan’s ambassador model ([83]).

In Serbia, public dissatisfaction centered on the COVID-19 Crisis Team—the government’s expert advisory body providing outbreak updates. The Crisis Team gradually lost public credibility as it imposed anti-epidemic measures unlawfully, bypassing the Government of Serbia. These measures impacted not only public health but also the economy and media ([38]). Political influence on the Crisis Team’s work may have contributed to the spread of misinformation and lack of transparency regarding COVID-19 death tolls ([38]), potentially linking to this study’s results.

This study’s findings on trust disparities and mental health outcomes during COVID-19 align with broader regional patterns in the Balkans and Eastern Europe, though contextual differences in media systems and institutional trust shape variations. When it comes to regional distrust drivers, post-communist distrust in institutions, exacerbated by corruption scandals (e.g., Bulgaria’s ventilator procurement fraud), underpins skepticism in public health measures ([72]).

### 4.5. Population of Students

Regarding students, elevated prevalence of anxiety and depression among higher education students during COVID-19 has been documented ([28]; [36]). Recent findings identified several correlates of heightened anxiety and depression levels among students during the pandemic, including recent traumatic events, pandemic-related financial concerns, remote learning, poor sleep quality ([36]), and increased academic frustration due to concerns about educational progress and quality ([3]).

In our study, college students emerged as the most vulnerable group, with significantly higher anxiety and depression scores compared to non-students. This aligns with studies from Sweden and India, where academic disruption, financial insecurity, and social isolation disproportionately affected students ([2]; [32]). Additionally, a higher degree of influence of the COVID-19 outbreak on depression levels in our study was observed among students. The level of decreased trust in preventive measures proposed by the Crisis team in the post-vaccination period was also significantly higher in students compared to non-students. These findings may be a product of different societal occurrences.

Specifically, young people in Serbia view most institutions as illegitimate, characterized by strong mistrust. The strongest mistrust is directed toward political parties: over two-thirds of young people distrust them, half distrust them completely, and only 4% express partial or complete trust. The only two institutions rated more trustworthy than untrustworthy are the military and the church ([55]). Recent research indicates slightly higher trust in healthcare among Serbian youth compared to political institutions and media, though this remains low relative to average trust levels across Serbian institutions ([67]). During the COVID-19 outbreak, young people perceived Serbian Government measures as contradictory and misaligned with citizens’ interests, viewing them as politically motivated ([7]).

The study’s finding of heightened anxiety and distrust among students aligns with global youth research. Karlsson et al. found Swedish students’ vaccine trust was heavily influenced by social media misinformation, amplifying fears of side effects and institutional neglect ([29]). In India, Kumar et al. noted that urban students distrusted vaccines due to digital rumors, while rural students faced access barriers, demonstrating how media and infrastructure jointly shape trust ([32]). Serbian students, exposed to similar digital misinformation, represent a global youth demographic disproportionately impacted by pandemic-related psychological stressors.

### 4.6. Healthcare Workers

The nature of job-related duties results in high anxiety levels among healthcare workers, especially during specific situations such as public health emergencies ([35]).

In our study, education and healthcare workers reported higher levels of anxiety and depression than military personnel. The impact of the COVID-19 outbreak on anxiety and depression levels was also more pronounced among education and healthcare workers compared to military personnel. However, significant differences in COVID-19-related psychological disturbance and public trust between these worker groups were not observed.

The rates of burnout, anxiety, and depression in healthcare workers climbed during the COVID-19 pandemic due to increased work hours, emotional demands, and emotional stress ([59]; [47]; [9]). Increased rates of burnout, symptoms of depression and anxiety, and increased social isolation and suicide rates are also predicted to climb both in the general population and within the population of U.S. physicians ([31]). Several previous findings confirmed increased depression and anxiety levels in healthcare workers, leading to a high risk of mental illness. One such study showed a major mental health disorder among nurses during the COVID-19 epidemic in an Iranian university ([56]).

Within our cohort of healthcare workers, medical nurses expressed a significantly lower level of trust in Serbia’s healthcare system compared to physicians. Also, the same trust decreased in nurses during the post-vaccination period, while it increased among medical doctors. This disparity mirrors global patterns where nurses, often undervalued and overburdened, exhibited higher vaccine hesitancy and burnout than physicians ([6]; [56]). Military personnel, conversely, reported the lowest psychological distress, likely due to structured environments and higher institutional trust—a phenomenon observed in nations with strong civic–military cohesion, such as Croatia and Japan ([71]; [83]). Therefore, as a part of the preparedness efforts of health care systems, continuous supervision of the psychological consequences following infectious disease outbreaks, especially among healthcare workers and workers in education sectors, should be undertaken by government institutions ([56]). Targeted interventions have been recommended, especially for those nurses who are directly involved with COVID-19 patients, highlighting some implications for mental health advocacy among nurses that are essential for improving the quality of services they provide and the safety of patients and staff in high-intensity conditions such as the COVID-19 pandemic. An additional reason for these recommendations is the fact that, compared to physicians, nurses expressed higher levels of anxiety during the COVID-19 pandemic ([60]; [76]). Also, it has been suggested that continuous supervision of the psychological consequences following infectious disease outbreaks should be a part of the preparedness efforts of healthcare systems ([62]; [42]; [68]). Organizational strategies focused on enhancing personal resilience and increasing social and organizational support in nurses have been suggested as well in order to reduce levels of anxiety related to public health emergencies, such as the COVID-19 pandemic ([33]).

### 4.7. Contribution of the Study to the Existing Theoretical Frameworks

This study provides critical insights into the interplay among media influence, trust in public health systems, and mental health outcomes during pandemics. Its findings contribute to existing theoretical frameworks—such as the Health Belief Model, Risk Perception Theory, and Extended Parallel Process Model—while offering empirical evidence of demographic disparities in trust and psychological distress. Below is a structured analysis of its contributions and implications.

The Health Belief Model posits that health behaviors are shaped by perceived susceptibility, severity, benefits, and barriers, as well as cues to action ([27]). The finding that physicians reported higher trust in the healthcare system than nurses aligns with the model’s emphasis on trust as a “cue to action.” Physicians, given their greater access to scientific information and institutional authority, likely perceived higher benefits of COVID-19 measures, mitigating their psychological distress. In contrast, nurses—frequently subject to systemic undervaluation and limited decision-making power—may have perceived higher barriers (e.g., workplace safety concerns), heightening distrust, and anxiety. This mirrors [6] ([6]), who found nurses globally exhibited higher vaccine hesitancy due to institutional mistrust.

Risk Perception Theory explains how individuals assess threats and take protective actions ([64]). The study demonstrates that COVID-19-related information amplified anxiety and depression, particularly among students, educators, and healthcare workers. This aligns with [44] ([44]), who showed that fear-driven media narratives (e.g., “pandemic chaos”) heighten risk perception and psychological distress unless accompanied by actionable solutions. Students, highly exposed to social media, likely encountered sensationalized or conflicting information, inflating perceived risk while lacking efficacy messaging—a dynamic addressed by Witte’s Extended Parallel Process Model ([78]).

Social Cognitive Theory emphasizes observational learning and self-efficacy within occupational contexts ([4]). The observed higher anxiety in education and healthcare workers (vs. military personnel) reflects their occupational exposure to pandemic stressors and societal expectations. Healthcare workers, despite clinical expertise, faced moral injury from resource shortages, undermining their self-efficacy. Conversely, military personnel, trained for crisis response, likely possessed stronger coping mechanisms. This parallels [40] ([40]), who linked occupational role conflicts to mental health declines in U.S. healthcare workers.

### 4.8. Implications for Policy and Practice

Institutional Reforms: Rebuilding trust requires transparent communication and anti-corruption measures. For example, Serbia’s Crisis Team, criticized for opaque decision-making ([38]), could adopt participatory frameworks involving healthcare workers and educators in policy design—a strategy that improved compliance in Germany and New Zealand ([48]).Mental Health Support: Targeted programs for high-risk groups are essential. Universities could integrate mental health screenings into academic services, while hospitals might offer resilience training for nurses. Japan’s “vaccine ambassador” model, which leveraged trusted community figures to boost uptake, offers a blueprint for Serbia ([83]).Media Regulation: Combating misinformation requires collaboration between governments and tech platforms. Serbia could emulate the EU’s Digital Services Act, which mandates transparency in algorithmic content prioritization, to curb viral conspiracy theories ([13]).Occupational Equity: Addressing nurse–physician trust disparities demands systemic changes, such as equitable resource allocation and leadership opportunities for nurses. Costa Rica’s nurse-led vaccination campaigns, which improved public confidence, highlight the value of empowering marginalized healthcare roles ([34]).

### 4.9. Study Limitations

Although this study had a cross-sectional design, its observational nature limits its ability to report specific causalities. Mediation was not formally analyzed due to sample constraints, and thus, the results reflect direct associations only.

Another limitation is the limited spectrum of information collected. In addition, the decreased ability to fully control the completeness of answers to the questionnaire’s questions provided by the participants resulted in a small degree of missing values, which, however, did not influence the validity of the results obtained. Also, BAI was used to assess the level of anxiety in participants, which measures anxiety symptoms but does not diagnose clinical disorders. Therefore, the results obtained related to the level of anxiety among participants may not reflect pathological forms of this condition. Additionally, the exclusion of mental health professionals may have omitted insights into clinical distress patterns.

The sampling method and online form of questionnaire are additional shortcomings, limiting this study’s potential to directly assess the predictive value of potential assessed factors on public trust, as well as anxiety and depression levels. Snowball sampling, as a non-probability method where existing participants recruit others, is prone to several biases that could compromise generalizability and validity, like selection bias, volunteer bias, social desirability bias, overrepresentation of close networks and urban, educated populations, as well as confirmation bias. Its disadvantages include a lack of representativeness, difficulty estimating sampling error, ethical risks, limited control over sample size/diversity, and reinforcement of existing structures. However, the snowball sampling method is useful in the current study, considering its exploratory characteristics and qualitative design.

Another major limitation of this study is the small size of subgroups and the absence of separate power calculations for each of the investigated groups, which emphasizes the importance of the cautious interpretation of results of between-group comparisons. Additionally, self-reported data risk recall and social desirability biases, though validated scales (BAI, SDS) mitigate the concern imposed by the sampling method used in this study. The study’s cross-sectional design precludes causal inferences; longitudinal studies are needed to track trust and mental health trajectories.

Finally, the level of distrust may also have been mis-represented, considering the fact that in authoritarian-leaning states, distrust may be underreported due to fear of repercussions.

## 5. Conclusions

Public trust and national media, as factors that can regulate the effects of pandemics, can increase perceived risk and appropriate safety behaviors, but also decrease psychological problems and disorders. Therefore, it is recommended that authorities consider critical issues and adopt appropriate approaches to prevent long-lasting negative effects on individuals and society in general.

However, the deficit or a decline in public trust in the healthcare system should not be considered solely a product of the sudden onset of medical emergencies and their characteristic of rapid progression, but also the ways governments generally treat citizens, economic standards of each country’s population, overall happiness of people, and their satisfaction by any life aspect directly or indirectly associated with the highest instances in individual countries.

Decreased public trust in the healthcare system observed among college students in Serbia in the post-vaccination era should not be observed only in the environment of the COVID-19 outbreak, but as a general position of the student population in Serbia. The uncertainty in the endangered future present among students compared to graduate workers serves as a good basis for future research. Therefore, it is critical to put more focus on the student population and to provide more flexible learning options and easier access to psychological services by universities and governments in order to ensure students’ safety as well as physical, social, and mental wellbeing.

This study illuminates the profound impact of institutional distrust and media polarization on mental health during COVID-19, particularly in Serbia’s post-vaccination era. By highlighting disparities among students, healthcare workers, and military personnel, it underscores the need for context-specific public health strategies. Rebuilding trust demands transparency, equity, and community engagement—principles that transcend pandemics and are vital for fostering resilience in future crises.

Future longitudinal studies are needed to track the evolution of trust in public institutions and its long-term impact on mental health outcomes across different societal groups, to develop and evaluate targeted interventions for high-risk populations in low-trust environments, as well as to explore cross-national comparisons to dissect how institutional structures, media ecosystems, and socio-political histories moderate the relationship among public trust, media exposure, and psychological distress during health crises.

Based on our findings, government agencies must work on enhancing transparent communication, funding targeted mental health programs, addressing occupational inequities, and combating misinformation. Actions conducted at universities could also significantly contribute towards improving mental health among students, especially in challenging circumstances, such as pandemic situations. These actions may include integrating mental health support into academic systems, promoting digital literacy programs, and creating structured communication channels. Healthcare institutions can also have a significant role in eliminating discrepancies in anxiety/depression levels, as well as public trust among healthcare workers by prioritizing nurse-specific interventions, promoting institutional transparency, and conducting resilience training. Community engagement initiatives, as a form of cross-sector collaboration, as well as national mental health task forces, would comprehensively fill the remaining gaps in stabilizing the overall mental health status of society, with a special focus on challenging circumstances. By adopting strategies, stakeholders can rebuild trust and resilience ahead of future health crises.

## Figures and Tables

**Figure 1 behavsci-15-00939-f001:**
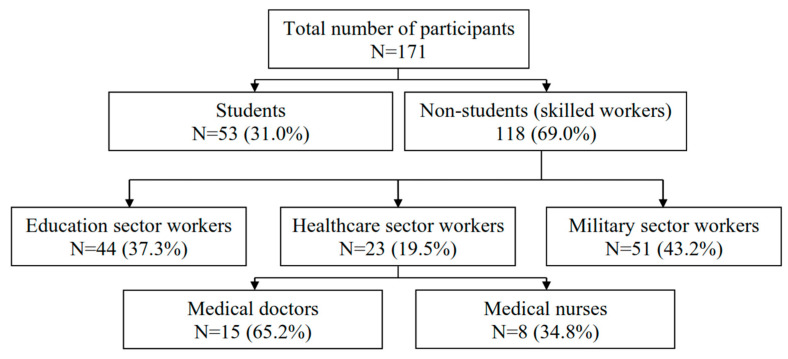
Distribution of the groups of participants.

**Figure 2 behavsci-15-00939-f002:**
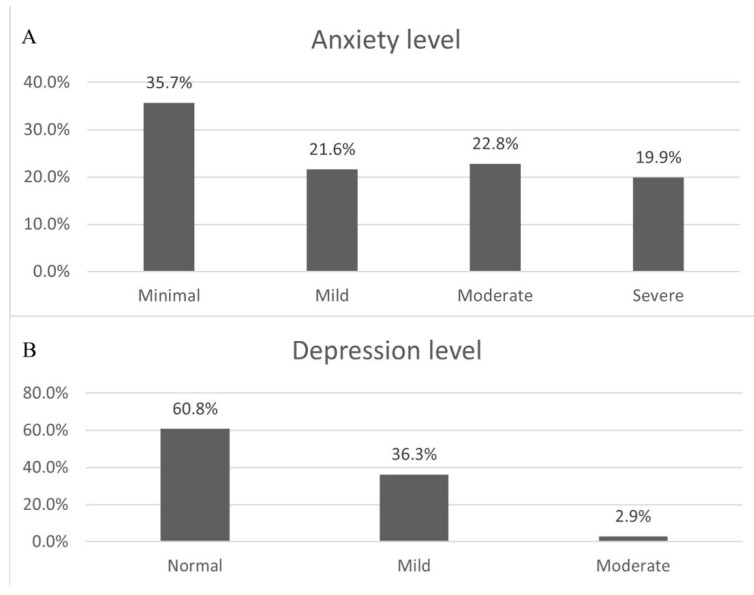
Distribution of (**A**) anxiety and (**B**) depression severity groups among participants.

**Figure 3 behavsci-15-00939-f003:**
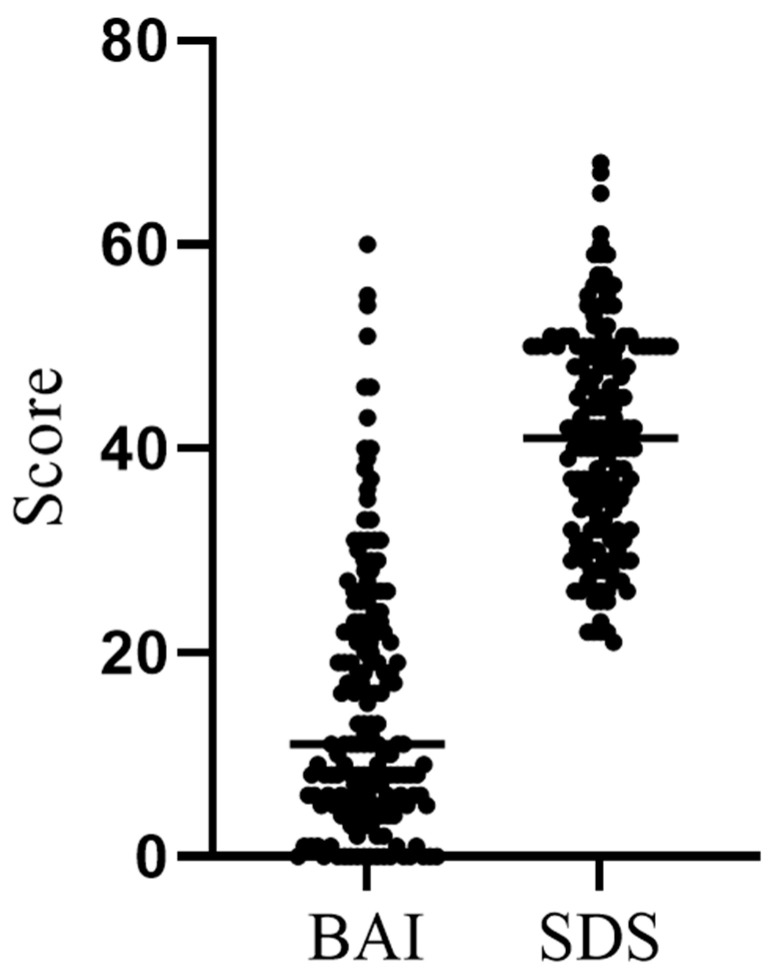
Distribution of individual anxiety and depression scores among participants (BAI—Beck Anxiety Inventory; SDS—Self-Rating Depression Scale).

**Figure 4 behavsci-15-00939-f004:**
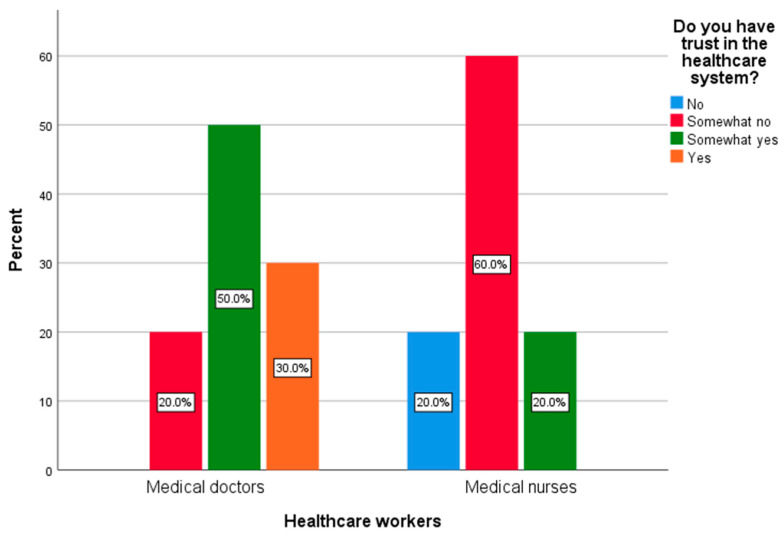
Levels of trust in healthcare system among medical doctors and medical nurses.

**Table 1 behavsci-15-00939-t001:** Frequency distribution of COVID-19-related origins of disturbance among participants.

Have You Experienced Disturbance	Never n (%)	Very Rarely n (%)	Rarely n (%)	Sometimes n (%)	Usually n (%)	Often n (%)	Very Often n (%)	Always n (%)
by the media reports regarding the outbreak?	44 (25.7)	39 (22.8)	22 (12.9)	32 (18.7)	14 (8.2)	7 (4.1)	8 (4.7)	5 (2.9)
by the information from other sources you have learned on your own initiative?	49 (28.7)	37 (21.6)	26 (15.2)	27 (15.8)	14 (8.2)	4 (2.3)	11 (6.4)	3 (1.8)
by the lack of the information regarding the COVID-19 outbreak and the disease itself?	49 (28.7)	36 (21.1)	28 (16.4)	29 (17)	8 (4.7)	11 (6.4)	6 (3.5)	4 (2.3)
by the possibility of virus transmission from other people despite personal preventive measures you applied?	48 (28.1)	28 (16.4)	26 (15.2)	35 (20.5)	10 (5.8)	9 (5.3)	12 (7)	3 (1.8)

COVID-19—Coronavirus Disease-19.

**Table 2 behavsci-15-00939-t002:** Change in COVID-19-related disturbance and public trust among participants compared to the period before the vaccination initiation.

Compared to the Period Prior to the Vaccination Implementation, What Was the Change in the Disturbance You Experienced	It Decreased n (%)	It Did Not Change n (%)	It Increased n (%)
by the media reports regarding the outbreak?	48 (28.1)	97 (56.7)	26 (15.2)
by the information from other sources you learned on your own initiative?	42 (24.6)	108 (63.2)	21 (12.3)
by the lack of the information regarding the COVID-19 outbreak and the disease itself?	38 (22.2)	115 (67.3)	18 (10.5)
by the possibility of virus transmission from other people despite personal preventive measures you applied?	45 (26.3)	112 (65.5)	14 (8.2)
**Compared to the period prior to the vaccination implementation, how did your trust change**	**It decreased** **n (%)**	**It did not change** **n (%)**	**It increased** **n (%)**
in the healthcare system	25 (14.6)	119 (69.6)	27 (15.8)
in the preventive measures proposed by the Crisis team	40 (23.4)	120 (70.2)	11 (6.4)

COVID-19—Coronavirus Disease-19.

**Table 3 behavsci-15-00939-t003:** Degree of influence of COVID-19 outbreak on BAI and SDS responses.

Answers n (%)	The Degree of the COVID-19 Outbreak Influence on the BAI Responses	The Degree of the COVID-19 Outbreak Influence on the SDS Responses
Smallest degree of influence	34 (19.9)	32 (18.7)
Very small degree of influence	32 (18.7)	39 (22.8)
Somewhat small degree of influence	45 (26.3)	46 (26.9)
Somewhat high degree of influence	43 (25.1)	36 (21.1)
Very high degree of influence	9 (5.3)	10 (5.8)
Highest degree of influence	8 (4.7)	8 (4.7)

COVID-19—Coronavirus Disease-19; BAI—Beck Anxiety Inventory; SDS—Self-Rating Depression Scale.

## Data Availability

The raw data supporting the conclusions of this article will be made available by the authors on request.

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
