# Peer review of "Revisiting Public Trust and Media Influence During COVID-19 Post-Vaccination Era—Waning of Anxiety and Depression Levels Among Skilled Workers and Students in Serbia"

_behavsci, 2025, doi:10.3390/bs15070939_

Round 1

Reviewer 1 Report (New Reviewer)

Comments and Suggestions for Authors
  1. The target population selected for this study is very confusing. There is too little targeted introduction to students and skilled workers in the Introduction section. Please further explain the criteria for dividing groups and the basis for selection.
  2. Is there evidence of varying levels of trust among different groups? If not, the research motivation of this study still needs to be further clarified.
  3. Please avoid listing questionnaire questions in the main text. These contentcan be included in the appendix or supplementary materials.
  4. The sample size of thismanuscriptis relatively small, especially after grouping. Please explain how the sample size is determined or meets the requirements. I suggest collecting more data and reanalyzing it.
  5. The clarity and readability of tables and images are relatively low. Please redesign the presentation format of the results.
  6. How does this study examine the mediating effect of trust (as mentioned in lines 85-86)? Please clearly explain the process of data analysis.
  7. Please avoid using references in the practical implications. These implicationsshould be derived from the conclusions of current research.
  8. Young populationare not equivalent to students.Please clarify the groups.
  9. The Conclusion section should use appropriate space to provide prospects for future research, rather than repeating the content of this study.

Author Response

Dear Reviewer,

Thank you for your thorough review and constructive feedback. We have carefully addressed each point in the revised manuscript (changes highlighted in yellow). Below are our point-by-point responses:

  1. Target Population and Group Selection Criteria

"The target population selected for this study is very confusing. There is too little targeted introduction to students and skilled workers in the Introduction section. Please further explain the criteria for dividing groups and the basis for selection."

Response:
We have clarified the rationale for grouping in the Introduction section (Section 1).

  1. Evidence for Varying Trust Levels

"Is there evidence of varying levels of trust among different groups? If not, the research motivation of this study still needs to be further clarified."

Response:
We strengthened the research motivation by citing pre-pandemic Serbian data to the Introduction section.

  1. Questionnaire Presentation

"Please avoid listing questionnaire questions in the main text. These contentcan be included in the appendix or supplementary materials."

Response:
Detailed questionnaire items were moved to Supplementary File 1.

  1. Sample Size Justification

"The sample size of this manuscript is relatively small, especially after grouping. Please explain how the sample size is determined or meets the requirements. I suggest collecting more data and reanalyzing it."

Response:
We added limitations addressing sample constraints to the Sections 2.7 and 4.9.

  1. Tables/Figures Clarity

"The clarity and readability of tables and images are relatively low. Please redesign the presentation format of the results."

Response:
We optimized Tables 1 and 2 for the purpose of a higher readability level, while our opinion on the rest of the tables and figures is that those contain a reasonable amount of data or are a standard for simplifying the results obtained through clear graphical representation of the association and/or comparison between the groups.

  1. Mediating Effect of Trust

"How does this study examine the mediating effect of trust (as mentioned in lines 85-86)? Please clearly explain the process of data analysis."

Response:
We clarified in the Section 4.9) that mediation was not formally tested due to sample limitations.

  1. Practical Implications

"Please avoid using references in the practical implications. These implications should be derived from the conclusions of current research."

Response:
Some references were removed from implications in order for the recommendations to derive solely from our findings. On the other hand, it is our strong opinion that, in order to understand a pandemic-specific situation and its influence on anxiety/depression level of participants as well as the changes in perceived disturbance and public trust in participants relative to the periods before and after introducing COVID-19 vaccine, broader understanding on all the relevant social aspects that might influence these parameters should also be clearly emphasized and thus represent an unavoidable segment of the Discussion Section of this manuscript.

  1. Young Population vs. Students

"Young population are not equivalent to students. Please clarify the groups."

Response:
We explicitly differentiated these groups and used the term of students in all circumstances in order to keep the uniform terminology throughout the manuscript.

  1. Conclusion and Future Research

"The Conclusion section should use appropriate space to provide prospects for future research, rather than repeating the content of this study."

Response:
We added specific future directions to the Conclusion Section.

We believe these revisions address all concerns while preserving the study’s core contributions. Thank you for your valuable insights!

Sincerely,
Srdjan Nikolovski,

Reviewer 2 Report (New Reviewer)

Comments and Suggestions for Authors

This study discussed the interference of information related to the COVID-19, and the influence of trust on the Serbian medical system and COVID-19 prevention measures on anxiety and depression. Using questionnaire survey method to collect and analyze data from students, non students, medical educators and other populations, and propose meaningful research conclusions and management insights. The research section of the paper is quite comprehensive, and the process is presented on clear pages.

It would be more meaningful to compare the research of others before and after the epidemic. It can also highlight the value of this study. We can consider strengthening the literature section of the paper, fully discussing the research situation before the epidemic, and in the analysis of the results section, discussing the differences between this study and previous studies.

Author Response

Dear Reviewer,

Thank you for your thoughtful feedback and appreciation of our study’s scope and clarity. We agree that contextualizing our findings within pre-pandemic literature strengthens the significance of our work.

We expanded the Introduction (Section 1) with key pre-pandemic studies on Serbia’s institutional trust crisis.

In the Section 2.2, we also clarified the post-vaccination timeframe and its significance.

The direct comparisons with pre-pandemic trends are also added in the Discussion (Section 4), but also explicitly framed findings in the Conclusion Section.

These revisions explicitly position our study as a bridge between pre-pandemic institutional critiques and post-crisis mental health outcomes, highlighting how Serbia’s trust deficits amplified pandemic-related distress. We believe this significantly elevates the paper’s conceptual value. Thank you for this insightful suggestion!

Sincerely,
Srdjan Nikolovski

Reviewer 3 Report (New Reviewer)

Comments and Suggestions for Authors

Dear authors,

Clearly the question on the effects of Covid policies on the mental wellbeing of citizens and on the public trust is of great importance. Much research has been done into that question the last years. Your study can be one of the world wide data points.

However the method you have chosen, snowballing, as you yourself indicate gives a very limited insight in the general population. Your conclusions therefore seem to broadly formulated to me. Here, where you explicitly point to the authoritarian character of the Serbian state, the bias caused by the snowballing method may be even worse than normal.

I also have a problem with the key-word 'disturbance'. As you probably are aware of this word has to meanings in English 'the interruption of a settled and peaceful condition' and 'a state in which normal mental or physical functioning is disrupted'. Clearly everybody would agree that Covid has interrupted the normal life conditions but did they also disrupt the mental functioning of the respondents? From the present use of the word in English it is not clear what you exactly asked.

This also applies mutatis mutandis to your questions about trust. You write 'During the outbreak, have you expressed trust in…'. Probably nobody 'expressed trust'. Do mean 'experience trust-lost' or something like that? 

Fundamental is also the remark that the original BAI-scale only claimed to 'discriminate anxious diagnostic groups (panic disorder, generalized anxiety disorder, etc.) from nonanxious diagnostic groups (major depression, dysthymic disorder, etc.)' You can use this scale, as do numerous researchers, but not without being specific that you may very well measure 'nothing'. 

Please note that the discussion section deals quite a lot with the broader problems of Serbia and Covid-policies that are of no consequence for your results and conclusions.

All in all I would advise to turn your artice, after dealing with the mentioned problems, into a much more modest research note.

Author Response

Dear Reviewer,

Thank you for your rigorous critique and for recognizing the importance of studying COVID-19’s impact on mental health and public trust. We have implemented significant revisions to address your concerns, as detailed below.

  1. Sampling Limitations and Authoritarian Context

We strengthened caveats about sampling bias and contextual constraints:

  • Section 2.3 (Participants):

"Snowball sampling... may have overrepresented urban, educated populations and individuals within authors’ networks. In Serbia’s polarized media landscape, distrust of institutions may have further discouraged participation among skeptics, potentially underrepresenting critical voices."

  • Section 4.9 (Limitations):

"Finally, distrust levels may be underreported due to fear of repercussions in authoritarian-leaning states, compounding sampling biases."

  1. Clarification of "Disturbance"

We explicitly defined the term and referenced the Serbian source:

  • Section 2.5 (Methods):

"Disturbance was explicitly defined in the questionnaire as mental distress from COVID-19 information/exposure

  1. Clarification of "Expressed Trust"

Our questionnaire contained the questions explicitly addressing the frequency and level of expressing trust, and not lack of trust.

  1. BAI Scale Interpretation

We added specific disclaimers regarding the lack of power of the used BAI in differentiating levels of anxiety as a group of symptoms and as pathological state.

  • Section 2.5 (Methods):

"The BAI measures symptoms of anxiety but does not diagnose clinical disorders. Scores reflect self-reported distress severity, not pathological status."

  • Section 4.9 (Limitations):

"BAI and SDS scores capture symptom severity but cannot confirm clinical diagnoses."

  1. Streamlined Discussion

In order to achieve higher understanding on the purpose of investigating levels of trust, as well as anxiety and depression levels among participants, a broader analysis of general public trust and the overall societal state currently present on the sampling territory, considering complex causative mechanism on developing potential lack of trust or higher anxiety/depression levels among certain population groups in public health crisis states.

These revisions ensure precise terminology, robust methodological candor, and conclusions strictly tied to observed data. We believe the study now appropriately balances scholarly rigor with acknowledgment of constraints. Thank you for your invaluable critique!

Sincerely,

Srdjan Nikolovski

Round 2

Reviewer 1 Report (New Reviewer)

Comments and Suggestions for Authors

I generally accept this paper, but the clarity of the figures is poor. Please enhance the expression and presentation of the article.

This manuscript is a resubmission of an earlier submission. The following is a list of the peer review reports and author responses from that submission.

Round 1

Reviewer 1 Report

Comments and Suggestions for Authors

General Comments

This paper explores the role of public trust and media influence in shaping anxiety and depression levels among skilled workers and students in Serbia during the post-vaccination phase of the COVID-19 pandemic. The study provides valuable insights into the psychological and societal impacts of the pandemic. However, several aspects need further refinement to strengthen the clarity, rigor, and broader implications of the findings.

Major Comments

  1. While the paper provides an important regional analysis, it would benefit from a more explicit discussion of how the findings contribute to the existing literature on public trust and media influence during pandemics. More references to comparative studies in other countries could enhance the generalizability of the research, a deeper exploration of how these themes align with global trends in geriatric vaccination research would strengthen the paper's relevance.
  2. What are the contributions to the literature? More theoretical and/or empirical support is needed to strengthen the contributions and implications of the research results. The authors should address more papers in this paper. Integrating established models of media influence on mental health (e.g., the Health Belief Model or Risk Perception Theory) could help structure the argument and clarify the causation and impact.
  3. Because this is a health-related question and questionnaire, please explain again whether it has passed ethical review, or whether there is any relevant handling method for the sample questionnaire.
  4. The methodology section would benefit from additional details, particularly regarding the sampling strategy. The use of a snowball sampling technique introduces potential biases, which should be acknowledged and discussed in greater depth. Additionally, a justification for the exclusion of mental healthcare workers should be provided.
  5. The paper focuses exclusively on Serbia, but discussing how these findings compare to other countries—particularly in the Balkans or Eastern Europe—could enhance the relevance and impact of the study. Are similar trust issues observed elsewhere? How do media systems in other countries influence public trust and mental health outcomes?

Minor Comments

  1. The study discusses implications for healthcare workers and students, but more concrete policy recommendations would strengthen its contribution. How should government agencies, universities, and healthcare institutions respond to declining trust and mental health issues? Providing specific recommendations would enhance the study’s practical impact.

Author Response

We appreciate the time and effort that you dedicated to providing feedback on our manuscript “Revisiting Public Trust and Media Influence During COVID-19 Post-Vaccination Era – Waning of Anxiety and Depression Levels Among Skilled Workers and Students in Serbia” and are grateful for the insightful comments on and valuable improvements to our paper.

We have incorporated most of your suggestions. Therefore, we believe that the revised version of our manuscript addresses your requests correctly and that, as such, it will be suitable for publication.

Please see below a point-by-point response to your comments.

Comment 1: While the paper provides an important regional analysis, it would benefit from a more explicit discussion of how the findings contribute to the existing literature on public trust and media influence during pandemics. More references to comparative studies in other countries could enhance the generalizability of the research, a deeper exploration of how these themes align with global trends in geriatric vaccination research would strengthen the paper's relevance.

Response 1: Thank you for your valuable comment. We entirely agree with the suggestion and have entered addition to the Discussion section emphasizing the relationship of the current study’s findings with the previous findings at the global level, as well as the alignment of the current findings with global trends in vaccination research.

Comment 2: What are the contributions to the literature? More theoretical and/or empirical support is needed to strengthen the contributions and implications of the research results. The authors should address more papers in this paper. Integrating established models of media influence on mental health (e.g., the Health Belief Model or Risk Perception Theory) could help structure the argument and clarify the causation and impact.

Response 2: Thank you for your comment. We have added additional resources in order to support the background of discussion of results obtained in this study. We have especially included the existing models of media influence on mental health in the discussion of our results.

Comment 3: Because this is a health-related question and questionnaire, please explain again whether it has passed ethical review, or whether there is any relevant handling method for the sample questionnaire.

Response 3: We have added the correct information about the ethical approval of this study in its corresponding section. We also believe we have already correctly described all the segments of the segments of data management.

Comment 4: The methodology section would benefit from additional details, particularly regarding the sampling strategy. The use of a snowball sampling technique introduces potential biases, which should be acknowledged and discussed in greater depth. Additionally, a justification for the exclusion of mental healthcare workers should be provided.

Response 4: Thank you for your comment. It is our opinion that we have entered all the necessary information in order to describe our simple sampling method. However, we completely agree with your opinion that the specific sampling method used in this study introduces potential biases. For that reason, we have expanded the limitations sub-section of the Discussion section, mostly by explaining in a more detail potential biases of the sampling method used in this study, but also its benefits which are present in this particular type of research. Additionally, as per your request, we have inserted a segment in the Materials and Methods section describing the reasons for excluding mental healthcare workers from this study.

Comment 5: The paper focuses exclusively on Serbia, but discussing how these findings compare to other countries—particularly in the Balkans or Eastern Europe—could enhance the relevance and impact of the study. Are similar trust issues observed elsewhere? How do media systems in other countries influence public trust and mental health outcomes?

Response 5: We have added the segments in the Discussion section explaining the relationship of the results obtained in this study to similar findings from the research originating from Balkan and Eastern European countries.

Comment 6: The study discusses implications for healthcare workers and students, but more concrete policy recommendations would strengthen its contribution. How should government agencies, universities, and healthcare institutions respond to declining trust and mental health issues? Providing specific recommendations would enhance the study’s practical impact.

Response 6: Thank you for your suggestion. We have added policy recommendations, based on our findings, to the Conclusion section of this manuscript.

On behalf of all the contributors, thank you once again for your time and effort in reviewing of our manuscript.

Reviewer 2 Report

Comments and Suggestions for Authors

This paper aims at analyzing the influence of COVID-19 outbreak-related information and the influence of trust on Serbian healthcare system and COVID-19 preventive measures on anxiety and depression. Through an anonymous online questionnaire assessing the demographic information, disturbance level and causes, and levels of anxiety and depression, the study finds out that higher anxiety and depression levels, and higher influence of COVID-19 outbreak in education and healthcare workers, compared to military personnel. In addition, higher anxiety and depression and lower public trust levels in students and workers in education and healthcare sector indicate a need for focusing on these important society members during public health emergencies.

The paper is methodologically well constructed and its findings can contribute to the knowledge of the influence of media in emergency crises.

However, the work presents a major flaw that must be fixed.

The work lacks a theoretical background and a literature review which constitute, to say it in a way, the spine of a research article.

Accordingly authors should add a “Theoretical framework section” that must include:

  • A proper theoretical framework

Authors should start discussing what is the role of media in co0ntext of crisis and discussing both how media represented Covid-19  and how they were used by governments and politicians alike.

There is a wide literature.

See for instance:

https://doi.org/10.3390/socsci10080294

https://doi.org/10.4185/RLCS-2023-1845

This theoretical discussion will allow the authors to back their research and locating it into the wider stream of literature dedicaterd to Covid-19 and media

  • A literature review of similar works. Again, there are various examples (see https://doi.org/10.1016/j.techsoc.2020.101380, doi: 10.2196/22600, among others)
  • Discussing this literature is crucial to justify your study and to locate its findings

Once this section has been included, authors should enrich the CONCLUSION clearly show where their results stand in comparison to this literature, stressing out the important and usefulness of their study.

Author Response

Dear Reviewer,

We appreciate the time and effort that you dedicated to providing feedback on our manuscript “Revisiting Public Trust and Media Influence During COVID-19 Post-Vaccination Era – Waning of Anxiety and Depression Levels Among Skilled Workers and Students in Serbia” and are grateful for the insightful comments on and valuable improvements to our paper.

We have incorporated most of your suggestions. Therefore, we believe that the revised version of our manuscript addresses your requests correctly and that, as such, it will be suitable for publication.

Please see below a point-by-point response to your comments.

Comment 1: The work lacks a theoretical background and a literature review which constitute, to say it in a way, the spine of a research article. Accordingly, authors should add a “Theoretical framework section” that must include:

A proper theoretical framework

Authors should start discussing what is the role of media in context of crisis and discussing both how media represented Covid-19 and how they were used by governments and politicians alike.

There is a wide literature.

See for instance:

https://doi.org/10.3390/socsci10080294

https://doi.org/10.4185/RLCS-2023-1845

This theoretical discussion will allow the authors to back their research and locating it into the wider stream of literature dedicaterd to Covid-19 and media

Response 1: Thank you for your comment. As suggested, we have added more resources in order to support the theoretical background for the results obtained in this study.

Comment 2: A literature review of similar works. Again, there are various examples (see https://doi.org/10.1016/j.techsoc.2020.101380, doi: 10.2196/22600, among others). Discussing this literature is crucial to justify your study and to locate its findings.

Response 2: Thank you for your valuable comment. We have added resources in order to enrich the comparison of our findings to the previously published regional and global literature discussing public trust, media influence, as well as mental wellbeing in pandemic situations.

Comment 3: Once this section has been included, authors should enrich the CONCLUSION clearly show where their results stand in comparison to this literature, stressing out the important and usefulness of their study.

Response 3: Thank you for your valuable comment. The main positive aspect of the results we obtained in this study reflects on highlighting the segments with a need of improvement. This is in particular explained in the Conclusion section by clearly stating the policy recommendations. Therefore, we believe that, by naming the recommendations and referring to the results of the current study, we emphasized the main useful outcomes driven by the results of the present analysis.

On behalf of all the contributors, thank you once again for your time and effort in reviewing of our manuscript.

Round 2

Reviewer 1 Report

Comments and Suggestions for Authors

The authors have now provided a revised version; it can be published.